# Recruitment and Aggregation Capacity of Tea Trees to Rhizosphere Soil Characteristic Bacteria Affects the Quality of Tea Leaves

**DOI:** 10.3390/plants13121686

**Published:** 2024-06-18

**Authors:** Xiaoli Jia, Shaoxiong Lin, Yuhua Wang, Qi Zhang, Miao Jia, Mingzhe Li, Yiling Chen, Pengyuan Cheng, Lei Hong, Ying Zhang, Jianghua Ye, Haibin Wang

**Affiliations:** 1College of Tea and Food, Wuyi University, Wuyishan 354300, China; jiaxl2010@126.com (X.J.); jhye1998@126.com (J.Y.); 2College of Life Science, Longyan University, Longyan 364012, China; 3College of JunCao Science and Ecology, Fujian Agriculture and Forestry University, Fuzhou 350002, China

**Keywords:** tea tree, rhizosphere soil, characteristic bacteria, nutrients, quality

## Abstract

There are obvious differences in quality between different varieties of the same plant, and it is not clear whether they can be effectively distinguished from each other from a bacterial point of view. In this study, 44 tea tree varieties (*Camellia sinensis*) were used to analyze the rhizosphere soil bacterial community using high-throughput sequencing technology, and five types of machine deep learning were used for modeling to obtain characteristic microorganisms that can effectively differentiate different varieties, and validation was performed. The relationship between characteristic microorganisms, soil nutrient transformation, and tea quality formation was further analyzed. It was found that 44 tea tree varieties were classified into two groups (group A and group B) and the characteristic bacteria that distinguished them came from 23 genera. Secondly, the content of rhizosphere soil available nutrients (available nitrogen, available phosphorus, and available potassium) and tea quality indexes (tea polyphenols, theanine, and caffeine) was significantly higher in group A than in group B. The classification result based on both was consistent with the above bacteria. This study provides a new insight and research methodology into the main reasons for the formation of quality differences among different varieties of the same plant.

## 1. Introduction

Organisms have memories, and previous perceptions that complex memory behaviors exist only in animals have been challenged by the introduction of plant memory behavior. Memory is the storage of information about past experiences by organisms and contains both permanent and stressful memories, and organisms perform memory behaviors on a daily basis. Higher animals store their memories in brain cells, whereas plants do not have any organized brain or nervous system; therefore, the existence of memory in plants is a topic of debate, but recent studies on the possible existence of memory in plants have been gradually unfolding [1]. Plants can accurately perceive and respond to environmental information through previous or parental experiences, and thus change their behavior to reconstruct similar stable systems [2]. For example, the immobility of plants limits some of their basic needs, and in order to meet their needs, plants rely on mnemonic behaviors to attract mobile organisms with specific chemical signals to help solve their needs [3]. Second, when faced with different needs, plants seek cues through memories formed from past experiences to initiate familiar pathways to address these needs [4]. For example, when plants face the first drought stress, although it causes some damage to plants, surviving plants will remember the coping strategies used in the first drought stress and form a genetic imprint memory, which will allow them to cope more calmly with the second drought stress [5]. This genetic imprint will allow the plant to be better adapted to the environment when it is not induced by any stress or stress signal. It will still transmit memory to its offspring in the form of genetic imprints, resulting in transgenerational memory inheritance, and the offspring will be better adapted to the environment [6]. Therefore, it is of great significance to explore the relationship between the native memory behavior of plants and their growth.

Soil is a seemingly lifeless but highly complex life complex that contains nutrients, plant roots, water, animals, and microorganisms that work together to support plant growth [7]. The vast majority of soil life is microorganisms, and these microorganisms regulate a myriad of soil chemical reactions that drive biogeochemical cycles, which in turn drive nutrient transformations in the soil and regulate plant growth [8,9]. Soil microbial communities are critical to plant health and productivity, and plant–microbe interactions promote rapid microbial evolution and altered function [10]. The interaction and co-evolution of plants and microorganisms over a long history has led to the formation of special memories of each other [11]. This mutualistic memory behavior is produced by the mutual drive between plants and microorganisms as they adapt to their environment [12]. Responding to memory behavior between plants and microorganisms, Kong et al. [13] proposed a new term “microbial-induced soil genetic” in 2019, which suggests that the microbial community in the rhizosphere zone of a plant is established at the time of the plant seed and is passed on from generation to generation, i.e., the seed begins to recruit and accumulate its memory microorganisms to safeguard plant growth. This behavior is the result of plants relying on partially persistent and moldable memories from seed germination to aggregate otherwise dispersed microbial communities and assemble dominant combinations of remembered microbes in response to current and possible future environmental changes [14], whereas microorganisms reorganize their functions after this particular aggregation, improving their defense against changes in environmental factors and being able to support plant growth more effectively [15]. It can be seen that there is a close correlation between the special ecological memory patterns formed between plants and soil microorganisms and plant growth. Ecological memory may better explain how plants influence the formation of microbial community structure, and how microbial communities, in turn, provide feedback to plants [16]. Ecological memory refers to the ability of organisms’ past states or experiences to influence their ability to cope with their current and future environments [17]. The fact that microorganisms are exposed to the same environment for a long period of time leads to the generation of memorized behaviors towards their environment, contributing to their relative stability in terms of community structure, diversity, and function [18]. For example, when plants are planted, their rhizosphere secretions selectively drive the microbial community structure in one direction, resulting in a mutual memory behavior between microbes and plants, which in turn affects plant growth [19,20,21,22,23]. Thus, it may be possible to distinguish plants from microbial ecological memory by identifying potentially aggregated microbial communities and their function in plant genetic imprints.

Tea tree (*Camellia sinensis*) is a plant in the Camelliaceae family, genus Camellia, which has an important economic value and plays an important role in promoting agriculture. The conservation of tea germplasm resources is a key foundation for the development of the tea industry [24]. All major tea-producing provinces in China have built tea germplasm resource nurseries to conserve tea germplasm resources. Tea tree germplasm resource nurseries are places where seedlings of wild or closely related tea tree varieties, originally grown in different regions and environments, are transplanted to the same plot at the same time and managed uniformly in a substantially similar environment [25]. It is well known that changes in environmental conditions affect the growth and quality of tea trees, and there is variability in this effect across tea tree varieties, especially on quality [26,27,28,29]. Numerous scholars have explained the differences in the quality of different tea tree varieties from genomic, transcriptomic, and genetic perspectives, which are mainly caused by differences in their physiological functions [30,31,32,33]. The internal mechanism of the tea tree itself explains that the tea tree has a genetic memory behavior, and this memory behavior shows some differences in quality. So how does the tea tree use its own memory to change its external environment to suit its growth? The authors hypothesized that another memory behavior may also exist in tea trees, and that this behavior would cause the ecological memory that exists between tea trees and microbes to prompt recruitment and aggregation of memory microbes in the tea tree rhizosphere to re-establish the memory microbial community structure after tea tree seedlings are transplanted. The reconstructed microbial community gradually affects nutrient cycling in rhizosphere soil, which in turn affects tea quality. Based on this hypothesis, in this study, we collected rhizosphere soils of 44 different tea tree varieties from the Tea Tree Germplasm Resource Nursery of Wuyi University, Wuyishan City, Fujian Province, China, and determined bacteria in rhizosphere soils of tea trees using 16S rDNA high-throughput sequencing technology. Unsupervised K-Means was used for the basic cluster analysis of soil bacteria, and then OPLS-DA combined with K-nearest neighbor (KNN), support vector machine (SVM), back propagation neural network (BPNN), random forest (RF), extreme gradient boosting (XGboost), and other machine deep learning techniques for modeling, was used to obtain the characteristic microorganisms distinguishing 44 tea tree varieties. Quantitative verification was performed using *q*RT-PCR. On this basis, the functions of the characteristic microorganisms were preliminarily described and their relationships with soil nutrients and tea quality were analyzed. In conclusion, the ecological memory behavior of tea trees and microorganisms determines the recruitment and aggregation capacity of characteristic microorganisms by the rhizosphere of tea trees, which in turn determines the transformation capacity of soil nutrients, and thus the quality of the tea leaves.

## 2. Results and Discussion

### 2.1. Basal Analysis of Rhizosphere Soil Bacterial OTUs of 44 Tea Tree Germplasm Resources

Miseq sequencing of tea tree rhizosphere soil bacteria showed that a total of 6,150,824 Mb of raw tags and 5,551,757 Mb of clean tags were obtained from 44 tea tree germplasm resources, with a Clean/Raw tag ratio of more than 85.20% (Appendix A), in which the sequence lengths of clean tags were mainly distributed between 200 and 540 bp (Appendix A). The obtained clean tags were further analyzed using clustering OTUs, and a total of 147,765 OTUs were obtained. The number of OTUs for each of the 44 tea germplasm resources was distributed between 2427 and 3961 (Appendix A), of which 252 OTUs were similar (Appendix A).

The rarefaction curve can be used to compare species abundance in samples with different amounts of sequencing data, and can also be used to indicate whether the amount of sequencing data in a sample is reasonable. When the rarefaction curve tends to be flat, it indicates that the amount of sequencing data is reasonable, and more data will only produce a small number of new OUTs [34]. The Shannon–Wiener index can reflect the microbial diversity of each sample at different sequencing amounts, and when the curve tends to be flat, it indicates that the amount of sequencing data is large enough to reflect the vast majority of microbial information in the sample [35]. The rarefaction curve, according to the Shannon–Wiener index of rhizosphere soil bacteria from 44 tea tree germplasm resources in this study flattened out (Figure 1A,B). As can be seen, the amount of sequencing data for test samples is reasonable, and the amount of sequencing data is large enough to reflect the vast majority of bacteria in the samples. Rank-abundance curves can be used to analyze species abundance and species evenness; the higher species abundance, the greater the range of the curve on the horizontal axis, and the flatter the curve, the more evenly species are distributed [36]. The rank-abundance curves of rhizosphere soil bacteria of 44 tea tree germplasm resources in this study had a wide range on the horizontal axis and steeper curves (Figure 1C). It can be seen that there were significant differences in the abundance of different bacteria and significant differences in species distribution in rhizosphere soil of 44 tea germplasm resources. Species accumulation curves can be used to describe the increase in species with the increase in sample size, and can be used to determine the adequacy of sample size, whereby the more appropriate sample size, the smoother the curve tends to be [37]. In this study, it was found that the species accumulation curve gradually leveled off as sample size increased (Figure 1D). It can be seen that 44 tea tree samples were sufficient for analysis, and even if additional samples were added, the increase in the number of bacterial species was minimal. Accordingly, the sample size taken in this study is reasonable, and the amount of sequencing data is large enough to reflect the vast majority of bacteria in the samples, and there are significant differences in abundance between different bacteria.

On this basis, this study further analyzed the rhizosphere soil bacterial diversity of different tea tree species. The results showed that the Simpson index of 44 tea tree germplasm resources was distributed between 0.97 and 1.00, with an average value of 0.99 (Figure 1E); the Shannon index was distributed between 8.07 and 9.88, with an average value of 9.37 (Figure 1F). It can be seen that the diversity of soil bacteria in the rhizosphere zone of different varieties of tea trees was rich. Second, further analysis revealed that the Chao1 index of rhizosphere soil bacteria of different tea tree varieties was distributed between 4149.15 and 6179.16, with an average value of 5461.62 (Figure 1G), and the PD whole tree index was distributed between 194.48 and 313.76, with an average value of 240.74 (Figure 1H). It can be seen that there were significant differences in the abundance of rhizosphere soil bacterial communities of 44 tea tree germplasm resources.

### 2.2. Rhizosphere Soil Bacterial Diversity Analysis of 44 Tea Tree Germplasm Resources

Based on the previous analysis, this study analyzed the bacterial diversity of the rhizosphere soil from 44 tea tree germplasm resources. Using the RDP classifier, the 147,765 OTUs obtained were classified into 745 phylotypes that could be assigned to members of the established bacterial genera and previously reported uncultured clones considered independent at genus level. Out of the 745 phylotypes, the following 16 accounted for more than 1% of the total abundance (Figure 2A): Ktedonobacteraceae *1921−2*, Ktedonobacteraceae *1921−3*, *Bryobacter*, *Acidibacter*, *Actinospica*, *Granulicella*, *Chujaibacter*, *Conexibacter*, *Acidothermus*, *Occallatibacter*, Ktedonobacteraceae *JG30a−KF−32*, *Bradyrhizobium*, Ktedonobacteraceae *HSB OF53−F07*, Candidatus *Solibacter*, *Acidobacteria bacterium*, and *Burkholderia-Caballeronia-Paraburkholderia*. It can be seen that although the tea tree varieties selected in this study were different, the dominant bacterial populations in their rhizosphere soils were extremely similar, as evidenced by the consistency of the bacterial genera, whose relative abundance accounted for more than 1% of the total. This result shows that tea trees have a certain memory behavior towards rhizosphere soil bacteria, and there is a strong similarity in the dominant bacterial populations recruited and aggregated by them.

The K-Means algorithm is an unsupervised learning algorithm that automatically assigns similar objects to the same cluster by continuously taking the closest mean to the seeded point, and is commonly used to discriminate samples of unknown categories [38]. In order to further analyze whether there are differences in the recruitment and aggregation capacity of rhizosphere soil bacteria among different tea tree germplasm resources, tea trees were further classified and analyzed from the perspective of bacterial communities. In this study, 44 tea tree germplasm resources were classified and analyzed using K-mean clustering based on the abundance of soil bacteria in the tea tree rhizosphere zone. The results showed (Figure 2B) that 44 tea tree germplasm resources could be effectively categorized into two groups—group A and group B-which included 31 and 13 tea varieties, respectively. Further, a bray–Curtis heat map analysis of the bacterial abundance of the two groups of tea tree varieties (Figure 2C) showed that there was a significant difference in bacterial abundance between group A and group B. It can be seen that different varieties of tea tree germplasm resources can be effectively distinguished from the perspective of bacterial community, which is manifested in 44 tea tree germplasm resources being distinguished into two groups.

### 2.3. Screening of Key Differential Bacteria between Group A and Group B

Screening key differential indexes between samples from different groups can be carried out by constructing the OPLS-DA model between sample groups to obtain the value of the importance projection (VIP value) of variables to screen key indexes [39]. After the model is constructed, the accuracy of the model must be analyzed using the permutation test to see if it reaches a significant level before subsequent analysis can be performed [40]. Accordingly, this study further constructed the OPLS-DA model for group A and group B based on rhizosphere soil bacterial abundance of 44 tea tree germplasm resources. The permutation test analysis showed (Figure 3A) that the OPLS-DA model had a goodness-of-fit R^2^Y value of 0.950 and a predictability Q^2^ value of 0.727 after 200 random simulations, both of which were highly significant levels (*p* < 0.005). It can be seen that the OPLS-DA model constructed in this study for group A and group B has a good fit and a high confidence for further analysis. Scores OPLS-DA plot analysis showed (Figure 3B) that the OPLS-DA model can effectively distinguish group A and group B in different coordinate areas. It can be seen that there is a significant difference in the rhizosphere soil bacterial community of tea trees between group A and group B. S-Plot analysis showed (Figure 3C,D) that the key differential bacteria between group A and group B belonged to 195 genera. Further, a Bray–Curtis heatmap analysis of key differential bacterial abundances between group A and group B (Figure 3E) showed that the difference in bacterial abundance between group A and group B was more pronounced. It can be seen that the rhizosphere soil dominant bacterial populations of tea trees between group A and group B are basically the same, while the key to distinguishing group A and group B lies in the abundance of bacterial communities. This result again suggests that there is a memory behavior of tea tree rhizosphere soil for bacterial communities, which is similar across tea tree varieties, but there is a difference in the ability of different tea tree varieties to recruit and aggregate bacteria, which is manifested as a significant difference between group A and group B in the abundance of the same bacteria.

### 2.4. Machine Deep Learning Validates 195 Key Bacterial Genera for Classification Accuracy and Screens to Obtain Characterized Bacterial Genera

Based on the above study, this study further employs five machine deep learning methods, such as KNN, SVM, BPNN, RF, and XGboost, to validate the accuracy of the classification of 195 key bacterial genera. The results showed (Figure 4A) that for validation using KNN, SVM, BPNN, RF, and XGboost, the classification accuracy between group A and group B was 0.94, 0.77, 0.94, 1.00, and 0.84 (ROC curves), respectively, and the overall accuracy was 96.00%, 88.00%, 99.88%, 97.12%, and 90.75% (confusion matrix). Among the five machine deep learning methods, the average classification accuracy of group A and group B was 0.90, and the average overall accuracy was 94.16%. It can be seen that the 195 key bacterial genera are able to effectively differentiate between group A and group B.

In machine learning, feature selection is a very important issue because the correct selection of features can improve the accuracy of the model, and feature importance is an index of the contribution of features in the model [41]. XGboost is a powerful machine learning algorithm that comes with the feature importance attribute that analyzes the importance of each feature in the model and identifies the features that are most important to the prediction results, thus improving the accuracy and interpretability of the model [42]. Random forest also has the ability to measure the importance of each feature, from which features that have a greater impact on the results are selected for further modeling to improve model accuracy [43]. Accordingly, this study used RF and XGboost to jointly assess the importance of 195 key bacterial genera in the classification. The results showed that the top 30 characteristic bacterial genera in terms of importance, obtained using RF, (Figure 4B) compared with the top 30 characteristic bacterial genera in terms of importance obtained using XGboost (Figure 4C), and there were 23 identical bacterial genera (Figure 4D), namely *Rhizomicrobium* sp., *Chujaibacter*, Candidatus *Solibacter*, *Bauldia*, *Acidobacteria bacterium*, *Verrucomicrobia bacterium*, *Hyphomicrobiaceae bacterium*, *Gemmatimonas*, *Fimbriimonas*, Candidatus *Udaeobacter*, Candidatus *Koribacter*, *Acidipila*, *Acidobacteriaceae bacterium*, *Chitinophaga*, metagenome, *Rhodoplanes*, *Bryobacter*, *Rhodospirillaceae bacterium*, *Bacterium Ellin5299*, Nitrosomonadaceae *MND1*, *Acidocella*, *Nitrolancea*, and *Haliangium*. Further, Bray–Curtis heatmap analysis of the abundance of 23 characteristic bacterial genera from group A and group B (Figure 4E) clearly showed that the difference in bacterial abundance between group A and group B was more pronounced. It can be seen that 23 characteristic bacterial genera may be the most important bacteria in distinguishing between group A and group B tea tree germplasm resources. At the same time, this result also illustrated that group A and group B, although extremely similar in their memory of rhizosphere soil bacteria, differ significantly in their ability to recruit and aggregate these 23 characteristic bacterial genera, as evidenced by significant differences in the abundance of the 23 genera of bacteria.

### 2.5. Validation of the Accuracy of Classification of 23 Characteristic Bacterial Genera and Their Abundance Analysis Using Machine Deep Learning

Based on 23 characteristic bacterial genera obtained, five machine deep learning methods such as KNN, SVM, BPNN, RF, and XGboost were used again in this study to validate the accuracy of its classification. The results showed (Figure 5A) that using KNN, SVM, BPNN, RF, and XGboost for validation, the classification accuracy of both group A and group B was 0.99, 0.97, 1.00, 0.97, and 0.88 (ROC curves), respectively, and the overall accuracy was 99.38%, 98.50%, 100%, and 93.12% (confusion matrix). The average classification accuracy of group A and group B of the five machine deep learning methods was 0.96, and the average overall accuracy was 97.90%. It can be seen that the accuracy of classification validation using the abundance of 23 characteristic bacterial genera for group A and group B was significantly improved. This result illustrated that the 23 characteristic bacterial genera were the most key bacteria for distinguishing between group A and group B. Accordingly, this study further analyzed the abundance of 23 characteristic bacterial genera in the rhizosphere soils of different tea tree germplasm resources from group A and group B. The results showed (Figure 5B) that 19 genera in the tea tree germplasm resources of group A had a relatively higher bacterial abundance than those of group B, while the opposite was true for the remaining four genera. Further significance analysis of the rhizosphere soil bacterial abundance from the germplasm resources of tea trees in group A and group B revealed that the bacterial abundance of 19 genera from group A was significantly higher than that of group B, while the opposite was true for the remaining four genera. It is evident that there was a marked difference in the recruitment and aggregation capacity of different tea tree varieties to rhizosphere soil bacteria, especially characteristic bacteria.

### 2.6. qRT-PCR Analysis and Functional Description of Characteristic Bacteria in the Rhizosphere Soil of Tea Trees

Based on the 23 characteristic bacterial genera obtained, and removing the unknown and those that could not be retrieved from the NCBI database for the time being, the remaining 18 genera of bacteria were further quantitatively analyzed using *q*RT-PCR in this study. The results showed (Figure 6A) that the number of bacteria from 15 genera was significantly higher in the rhizosphere soil of tea tree germplasm resources in group A than in group B, namely *Rhodospirillaceae bacterium*, Candidatus *Koribacter*, Candidatus *Solibacter*, Candidatus *Udaeobacter*, *Chitinophaga*, *Fimbriimonas*, *Gemmatimonas*, *Haliangium*, *Rhodoplanes*, *Acidobacteria bacterium*, *Acidobacteriaceae bacterium*, *Hyphomicrobiaceae bacterium*, *Rhizomicrobium* sp., *Verrucomicrobia bacterium*, and *Bryobacter*. In contrast, group B had three genera with significantly higher bacterial numbers than group A, namely *Nitrolancea*, *Acidocella*, and *Chujaibacter*. This result was consistent with the trend of characteristic bacterial abundance changes in the high-throughput sequencing described above, which fully validated the previous results. In addition, based on the *q*RT-PCR results, this study continued to re-validate the classification accuracy of group A and group B using five machine deep learning methods such as KNN, SVM, BPNN, RF, and XGboost, as well as Bray–Curtis heat map analysis. The results showed (Figure 6B) that the difference in bacterial abundance between group A and group B was more pronounced, as seen on the Bray–Curtis heat map, and this confirmed the above conclusion. Second, the results of the five machine deep learning methods showed (Figure 6C) that the average classification accuracy of group A and group B was 0.97 (ROC curve), and the average overall accuracy was 98.72% (confusion matrix). This result also reaffirmed the conclusion that the above-mentioned characteristic bacteria could effectively differentiate between group A and group B.

Accordingly, this study further provided a preliminary description of the functions of 18 characteristic bacterial genera based on the available literature. *Candidatus Udaeobacter* has been reported to be a functional bacterium with the ability to secrete antibiotics and remove potential trace gases, which are beneficial for improving soil texture [44]. *Acidobacteria bacterium* has a positive contribution to enhancing soil biological activity [45]. *Rhizomicrobium* sp. is related to soil texture, and soil degradation can lead to a decline in the number of *Rhizomicrobium* sp. [46]. In this study, it was found that the number of Candidatus *Udaeobacter*, *Acidobacteria bacterium*, and *Rhizomicrobium* sp. was significantly higher in the rhizosphere soils of group A tea tree germplasm resources than group B. It can be hypothesized that the tea tree germplasm resources of group A may be more conducive to the recruitment and aggregation of Candidatus *Udaeobacter*, *Acidobacteria bacterium*, and *Rhizomicrobium* sp. in the soil, which in turn was more conducive to the improvement of soil texture.

*Gemmatimonas* was reported to be an important microorganism that responds to changes in soil nutrient content, and its abundance increased with higher soil fertility [47]. *Acidobacteriaceae bacterium*, as a key microorganism in soil fertility assessment, is beneficial to improving soil and enhancing soil fertility [48]. *Fimbriimonas* is significantly correlated with soil fertility and increasing its abundance can significantly increase the available nutrient content of the soil [49]. In this study, the number of *Gemmatimonas*, *Acidobacteriaceae bacterium*, and *Fimbriimonas* was significantly higher in the rhizosphere soils of the tea tree germplasm resources of group A than in group B. It can be hypothesized that group A tea tree germplasm resources were more conducive to the recruitment and aggregation of such bacteria in the soil, which in turn improved soil fertility and promoted tea tree growth.

*Rhodospirillaceae bacterium* and *Chitinophaga* have been reported to play an important role in soil C cycling and fixation, which is conducive to improving soil C cycling capacity and accelerating soil C metabolism [50,51]. *Haliangium* facilitates soil carbon cycling, accelerates the decomposition of organic matter, and provides nutrients to the soil [52]. *Verrucomicrobia bacterium* favors the improvement of soil texture, increases the available nutrient content, and promotes plant growth [53]. *Hyphomicrobiaceae bacterium* can promote C and N cycling in soil, improve soil texture, and increase soil nutrient content [54]. *Rhodoplanes* are involved in nitrogen transformation and can increase the content of available nitrogen in the soil [55]. *Bryobacter* regulates the content and effectiveness of P and K in the soil, which in turn improves plant yield and quality [56]. Candidatus *Koribacter* and Candidatus *Solibacter* improve N and P cycling in soil, and increase available nitrogen and phosphorus content in soil [57]. In this study, it was also found that the number of all these bacteria was significantly higher in the rhizosphere soil of group A tea tree germplasm resources than in group B. It can be seen that group A tea tree germplasm resources were more conducive to the recruitment and aggregation of such bacteria in the soil, which in turn increased the content of soil available nutrients and improved soil fertility.

Secondly, it was also found in this study that the number of *Nitrolancea*, *Chujaibacter*, and *Acidocella* in the rhizosphere soil of group A tea tree germplasm resources was significantly lower than that in group B. *Nitrolancea* correlated with the rate of nitrogen conversion in the soil, increasing the rate of soil nitrification and accelerating nitrogen loss [58]. *Chujaibacter* abundance is significantly and positively correlated with soil organic oxygen compounds, and increasing its abundance reduces the content of soil available nutrients [59]. *Acidocella* facilitates the degradation of soil hazardous substances and improves soil texture [60]. It can be seen that from the overall perspective, there was a significant difference in the recruitment and aggregation ability of group A and group B for the 18 characteristic bacterial genera, while in terms of the functions of the characteristic bacteria and their numbers, the rhizosphere soil bacteria of group A tea tree germplasm resources were more conducive to promoting soil nutrient cycling.

In summary, through the preliminary description of the functions of the 18 characteristic bacterial genera, this study hypothesized that there was similarity in the ecological memory of rhizosphere soil bacteria of tea tree germplasm resources in group A and group B, but there was a significant difference in the ability to recruit and aggregate the characteristic bacteria. Group A rhizosphere soils of tea tree germplasm resources were significantly stronger in recruiting and aggregating 15 characteristic bacteria and weaker for the other three characteristic bacteria. In terms of function and number of characteristic bacteria, the characteristic bacteria in rhizosphere soil of group A tea tree germplasm resources were more conducive to improving soil texture and nutrient cycling capacity of the soil, which in turn increased the content of available nutrients in the soil, and thus could be more conducive to promoting growth and improving the quality of group A tea tree germplasm resources.

### 2.7. Analysis of the Content of Available Nutrients in the Rhizosphere Soil of Tea Tree

Based on the above speculation, this study further determined the available nutrient contents of the rhizosphere soils of 44 tea tree germplasm resources, and the results showed (Figure 7) that the available nitrogen contents of the rhizosphere soils of group A tea tree germplasm resources were distributed in the ranges of 30.52~96.38 mg/kg, the available phosphorus contents were distributed in the ranges of 4.95~12.03 mg/kg, and the available potassium contents were distributed in the ranges of 60.25~107.32 mg/kg. The contents of available nitrogen, available phosphorus, and available potassium in the rhizosphere soils of group B tea tree germplasm resources were distributed in the ranges of 20.19~40.08 mg/kg, 3.49~6.02 mg/kg, and 30.24~68.13 mg/kg, respectively. Significance analysis showed (Figure 7) that the content of available nutrients (N, P, and K) in the rhizosphere soil of tea tree germplasm resources of group A was significantly higher than those of group B. The results validated the conclusions hypothesized after the above functional description of the characteristic microorganisms. Secondly, this study attempted to classify 44 tea tree germplasm resources using five machine deep learning methods based on soil available nutrient contents. The results showed (Appendix A) that the 44 tea tree germplasm resources were still classified into two groups, and their classification results were consistent with those of the above classification of bacterial communities, with an average accuracy of 0.99 (ROC curve) and an average overall accuracy of 99.65% (confusion matrix). It is evident that there was a close relationship between the recruitment and aggregation capacity of rhizosphere soil microorganisms by different tea tree varieties and the content of available nutrients of the soil.

It has been reported that the available nutrient content of the soil has a close relationship with the quality of tea, and increasing the available nutrient content of the soil is conducive to improving tea quality [61,62,63,64]. The improvement of tea quality is of great significance to the enhancement of tea’s economic benefits, and this study found that the available nutrient content in the rhizosphere soil of the tea tree germplasm resources of group A was significantly higher than that of group B. Does this phenomenon lead to certain differences in tea quality between the two groups as well?

### 2.8. Content Analysis of Tea Quality Indexes

Tea polyphenols, theanine, and caffeine are important indexes for evaluating tea quality, and many scholars often use the content of these three indexes to evaluate tea quality, with a high content of good quality and vice versa [65,66,67]. Accordingly, on the basis of the above speculation, this study further determined the quality index content of 44 tea tree germplasm resources. The results showed that (Figure 8) the tea polyphenol, theanine, and caffeine contents of the tea germplasm resources of group A varied in the ranges of 30.31~36.90 mg/kg, 2.94~4.73 mg/kg, and 3.02~5.40 mg/kg, respectively; while the variation ranges of these indexes of group B tea tree germplasm resources were 23.60~28.93 mg/kg, 2.42~2.91 mg/kg, and 2.75~3.13 mg/kg, respectively. Significance analysis showed (Figure 8) that all three quality indexes in the leaves of group A tea tree germplasm resources were significantly higher than those in group B. This result validated the conclusions hypothesized in the previous description. In addition, this study attempted to classify 44 tea tree germplasm resources using five machine deep learning methods based on the tea quality index content. The results showed (Appendix A) that the 44 tea tree germplasm resources were still classified into two groups, and their classification results were consistent with those of the bacterial community and available nutrient content in the previous section, with the average classification accuracy reaching 0.88 (ROC curve) and the average overall accuracy reaching 89.77% (confusion matrix). It can be seen that group A and group B did differ significantly in tea quality, which resulted from their ability to recruit and aggregate characteristic soil bacteria that altered the content of available nutrients in their rhizosphere soils, and hence their quality.

### 2.9. Interaction Analysis between Characteristic Bacteria, Soil Available Nutrients, and Tea Quality Indexes

Based on the above study, this study further analyzed the interactions of 18 characteristic bacteria with soil available nutrients and tea quality indexes. Redundancy analysis (RDA) showed (Figure 9A) that there were 15 characteristic bacteria significantly correlated with group A, and 3 characteristic bacteria significantly correlated with group B. This analysis also showed that soil available nutrients (N, P, and K) and quality indexes (tea polyphenol, theanine, and caffeine) were significantly correlated with group A. Interaction network analysis (Figure 9B) showed that the 15 characteristic bacteria significantly associated with group A were significantly positively correlated with both soil available nutrients and tea quality indexes, while the 3 characteristic bacteria significantly associated with group B were significantly negatively correlated with both soil available nutrients and tea quality indexes. Partial least square structural equation model (PLS-SEM) analysis showed (Figure 9C) that from a bacterial point of view, 15 characteristic bacteria had a positive effect on the bacterial number, and the other 3 characteristic bacteria showed the opposite effect, while 18 characteristic bacteria had a positive effect on the content of available nutrients in the soil and reached a highly significant level (*p* < 0.001). Secondly, the effect of soil available nutrient content on the quality index content of tea leaves was positive and highly significant (*p* < 0.001). It can be seen that the memory behaviors of 44 tea tree germplasm resources towards rhizosphere soil bacteria led to some similarities in the recruitment and aggregation of bacterial communities, but differences in recruitment and aggregation abilities led to some differences in quantity, which in turn led to differences in function. The ability of rhizosphere recruitment and aggregation of 15 characteristic bacteria of group A tea tree germplasm resources was significantly higher than that of group B, which was more conducive to the increase of available nutrient content in the soil, and more conducive to the improvement of tea quality. It is evident that the tea tree’s ability to recruit and aggregate characteristic bacteria in the rhizosphere determines the quality of tea leaves.

## 3. Materials and Methods

### 3.1. Experimental Design and Sample Collection

The experimental sites of this study were located in the Tea Tree Germplasm Resource Nursery of Wuyi University, Wuyishan City, Fujian Province, China (117°59′51.88″ E, 27°44′16.17″ N), and the age of all the tea trees was 10 years. The experimental site was located at an altitude of 216 m, with an average annual temperature of 19 °C, an annual rainfall of 1284 mm, and an average humidity of 74%. In October 2022, the rhizosphere soils and leaves of 44 tea tree varieties (Table 1 and Figure 10) were collected, of which soil was used to determine available nutrient content; amplicon sequencing analysis of bacterial diversity and subsequent *q*RT-PCR analysis of important characteristic microorganisms and leaves were used to determine tea quality indexes. The rhizosphere soil of the tea tree adopted an S sampling method; that is, five tea trees were randomly selected, the fallen leaves on the soil surface were removed, the tea tree was gently dug out, the root was removed, the soil adhering to the root surface was shaken off, and the soil still adhering to the root was collected as the rhizosphere soil of the tea tree, which was then appropriately mixed to be a replicate. Three independent replicates were taken for each tea tree variety. The sampling method for tea tree leaves was to collect the leaves of the corresponding tea trees while collecting the rhizosphere soil of the tea trees, i.e., one bud and two leaves of five randomly selected tea trees, and mixing them thoroughly to make a replicate. Three independent replicates were taken for each tea tree variety.

### 3.2. 16S rDNA Amplicon Sequencing Analysis

DNA extraction: Rhizosphere soil microbial DNA from tea trees was extracted using instructions from the Bio-Fast Soil Genomic DNA Extraction Kit (BioFlux, Hangzhou, China). The extracted DNA was detected using 1% agarose gel electrophoresis, and the DNA was purified using the gel recovery kit of TianGen Biotech Co., Ltd., Beijing, China. DNA concentration was determined using UV spectrophotometry and was used for PCR amplification after quality control.

PCR amplification and sequencing: 16S rDNA amplification of soil bacteria referred to the method of Zhang et al. [52]. PCR amplification primers were 338F (ACTCCTACGGGGAGGCAGCAG) and 806R (GGACTACHVGGGTWTCTAAT). The PCR reaction system was 12.5 μL 2xTaq Plus Master Mix, 3 μL BSA (2 ng/μL), 1 μL forward primer (5 μM), 1 μL reverse primer (5 μM), 2 μL DNA (total amount of DNA added was 30 ng), and 5.5 μL dd H_2_O. The PCR reaction program was pre-denaturation at 94 °C for 5 min; denaturation at 94 °C for 30 s; annealing at 50 °C for 30 s; extension at 72 °C for 60 s, 30 cycles; and extension at 72 °C for 7 min. PCR products were purified using Agencourt AMPure XP Nucleic Acid Purification Kit (VWR, Radnor, PA, USA) and were used to construct microbial diversity sequencing libraries. After library construction, paired-end sequencing was performed using the Illumina Miseq PE300 high-throughput sequencing platform at Beijing Allwegene Technology Co., Ltd., Beijing, China.

Bioinformatics analysis: Purified amplicons representing the bacterial 16S rDNA gene sequence reads were performed using Illumina Analysis Pipeline Version 2.6. First, the Fastq data were analyzed for quality control using trimmomatic (v0.36) and pear (v0.9.6), where trimmomatic used a sliding window strategy with the window size set at 50 bp, the average quality value of 20, and the minimum retained sequence length of 120 bp, and pear was used to remove sequences with N. According to the paired-end overlap relationship, Flash (v1.20) and pear software (v0.9.6) were used to splice the two-end sequences, with a minimum overlap setting of 10 bp and a mismatch rate of 0.1, thus obtaining sequences in Fasta format for raw tags. Based on this, the uchime method was further used to compare and remove chimeras from Fasta sequences and non-compliant short sequences as a way to obtain clean tags. The purified sequences were clustered into operational taxonomic units (OTUs) at a 97% similarity level against the SILVA v128 reference database set at a minimum support threshold of 70%. The Ribosomal Database Project (RDP) classifier tool was used to classify all sequences into different taxonomic groups. The rarefaction curve, Shannon–Wiener, rank-abundance, and species accumulation curves were produced for the obtained data. A random sampling method was used to construct a rarefaction curve with the number of sequences selected versus the number of OTUs they could represent. A Shannon–Wiener curve was constructed using the bacterial diversity index at different sequencing depths for each sample, as a reflection of the microbial diversity of each sample at different sequencing quantities. The abundance of OTUs was sorted from large to small rank, and then rank-abundance curves were produced using the OTU rank as the horizontal coordinate and the number of sequences contained in each OTU as the vertical coordinate. Species accumulation curves were produced using the sample size as the horizontal coordinate and the rate of emergence of new OTUs under continuous sampling as the vertical coordinate. A diversity analysis such as Shannon, Simpson, Chao1, and PD whole tree was performed using QIIME1 (v1.8.0) software.

### 3.3. Construction and Evaluation of Machine Learning Models

Machine learning algorithms are widely used to sift and categorize big data, but so far there is no single algorithm that works in all situations. Therefore, using multiple algorithms for modeling leads to better construction of high-quality models. In this research, five classical machine learning algorithms, including K-nearest neighbor (KNN), support vector machine (SVM), back propagation neural network (BPNN), random forest (RF), and extreme gradient boosting (XGboost), were used to analyze and model rhizosphere soil bacteria from 44 tea tree germplasm resources (Figure 11). All model construction was based on R software version 4.2.3, and evaluation parameter tables were produced in Excel. In the process of model construction, this study adopts a stratified sampling technique, that is, data samples are divided into training sets and test sets in the ratio of 80% and 20%. To further ensure the stability of the model, 100 randomized iterations of training were conducted in this study, and the training set and test set were randomly combined proportionally in each training iteration. Finally, the model integration analysis was carried out with the results of 100 iterations of training, which integrated all the models from the training process into a comprehensive result, improving the stability and accuracy of the model predictions.

The KNN classification algorithm is one of the simplest machine learning algorithms, based on the principle of determining attributes based on the category of the nearest k points; predicting new values is simple, fast, and insensitive to outliers. The choice of k value can have a significant impact on the results of the algorithm. When the k value is small, the overall complexity of the model decreases and it is easy to overfit. When the k value is large, it selects training set instances that are far from the validation set samples, which affects the prediction and makes the prediction inaccurate [68]. In practice, the k value is usually chosen to be smaller and then the optimal k value is selected using cross-validation. In this study, a tuning grid containing k values, where k values range from 1 to 20, is set up for cross-validation and parameter optimization.

The core idea of SVM is to find an optimal hyperplane that maximizes the interval between two categories. By introducing a kernel function, SVM is able to handle nonlinear classification problems [69]. In this study, a radial basis kernel function (RBF)-based SVM is used, and in order to obtain the optimal model parameters, a parameter grid containing C and σ values, where both C and σ values range from 10^−3^ to 10^3^, increasing by one order of magnitude at a time, is defined for cross-validation and parameter optimization.

The BPNN classification algorithm is also an effective machine learning algorithm. In BPNN, information is propagated forward through layer-to-layer connections and error backpropagation and weight updating is performed through a backpropagation algorithm to minimize the error between the predicted output and the actual output [70]. In this study, a neural network model of backpropagation with a single hidden layer is designed and constructed. The model has a hidden layer between the input and output layers, which is designed to capture and process complex features in the data, thus enhancing the model’s learning and representation capabilities.

RF is a comprehensive supervised classification algorithm that can be used to solve large errors and overfitting problems that may occur with individual decision trees. Second, RF performs well in classification problems and has a great potential to be the classifier that works best in each case [71]. The model consists of many decision trees that are unrelated to each other. This model, after obtaining the forest, when judging or predicting new samples, judges each decision tree in the forest separately to distinguish which category the sample belongs to and compares which category has the highest number of selections, thus making a judgment on the sample category and deciding how many trees should be in this model is crucial [72]. The number of trees in this study was initially set at 500 and a parameter grid containing mtry (number of variables considered per split), where mtry ranges from 1 to 59, was defined for cross-validation and parameter optimization. In addition, the importance scores of the impact of all variables on model construction were derived from the model constructed using RF in order to evaluate the value of the contribution of different variables to model construction and thus obtain the importance characteristic variables.

XGboost is an efficient, flexible, and portable gradient enhancement algorithm that has been widely used in various classification and regression tasks. XGboost uses an additive strategy to incrementally add trees and optimizes the objective function at each step using the gradient boosting principle. This allows the model to attempt to better correct the residuals of all previous trees at each step, resulting in more accurate predictions [73]. In order to find the optimal model parameters, this study defines a parameter grid with nrounds (number of augmentation iterations), max_depth (maximum depth of the tree), eta (learning rate), colsample_bytree (the ratio of subsamples of the features used in the construction of each tree), min_child_weight (the minimum weight of the child nodes), and subsample (the ratio of subsamples of the training data) for cross-validation and parameter optimization. In addition, the models constructed using XGboost were used to derive the importance scores of the impact of all variables on model construction in order to evaluate the value of the contribution of the different variables to model construction and thus obtain the importance characteristic variables.

After all the models were constructed, this study made a preliminary evaluation of the performance of the models through the confusion matrix and ROC curves. Confusion matrices are often used to evaluate the classification effect of each group, reflecting the relationship between the true category of sample data and the predicted results, and quantifying classification details more intuitively [74]. In this study, after completing the iteration of each machine learning algorithm, model integration was used to calculate the overall confusion matrix of the model and draw a heat map of the confusion matrix. Model integration can be seen as a strategy that combines the predictions of multiple models to produce a more robust and accurate combined prediction [75]. With this approach, the confusion matrix for the model as a whole can be computed to obtain an overall assessment of the predictive performance of all models. To further deepen the understanding of model performance, this study plots the overall ROC curve after completing the iteration of each machine learning algorithm. ROC curves are a common tool for evaluating the performance of binary classifiers, which provide a comprehensive view of the model’s performance by plotting the relationship between true positive and false positive rates at different classification thresholds. The generalization ability of the model is also evaluated by the area under the curve (AUC) in the ROC curve. The AUC value ranges from 0 to 1.0 and is positively correlated with model quality [76,77].

### 3.4. qRT-PCR Analysis of Soil Characteristic Bacteria

Based on the analysis of the research data in this study, 23 characteristic bacterial genera were obtained. Bacteria that were unknown and temporarily unsearchable in the NCBI database were removed, and the number of bacteria in 18 of these genera was quantitatively analyzed in this study. Briefly, soil DNA was extracted using the Bio-Fast Soil Genomic DNA Extraction Kit (BioFlux, Hangzhou, China), and DNA was detected using 1% agarose gel electrophoresis and was purified using the gel recovery kit of TianGen Biotech Co., Ltd., Beijing, China, to perform *q*RT-PCR analysis. Fluorescence quantitative PCR curves were performed with different concentrations of plasmids, and copy numbers of microorganisms amplified in different samples were determined and converted to quantitative data by using a real-time fluorescence quantitative PCR system of ABI7500 (Thermo Fisher, Carlsbad, CA, USA). The *q*RT-PCR primers were designed according to the conserved 16S rDNA sequences from each of the 18 bacterial genera and were then analyzed using PCR, as shown in Appendix A. The PCR programs were all set to 94 °C pre-denaturation for 5 min, 94 °C for 15 s, 60 °C for 30 s, 72 °C for 15 s, and 30 cycles.

### 3.5. Determination of Available Nutrient Content of Rhizosphere Soils of Tea Trees

Rhizosphere soils collected from 44 different tea tree varieties were used to determine the available nitrogen, available phosphorus, and available potassium content of the soils, as described in Wang et al. [78]. The available nitrogen content was determined using alkalolytic diffusion, i.e., the soil was extracted by NaOH solution (1 mol/L), filtered, and the filtrate was then titrated with hydrochloric acid, which was converted to obtain the available nitrogen content of the soil. The available phosphorus content was determined using molybdenum–antimony resistance colorimetry, i.e., the soil was leached with NaHCO_3_ solution (0.5 mol/L), filtered, the filtrate was added to molybdenum–antimony resistance color reagent, and then the absorbance was measured at 880 nm, which was converted to obtain the available phosphorus content of the soil. The available potassium content was determined using flame photometry, i.e., the soil was leached with neutral ammonium acetate (1 mol/L), filtered, and the filtrate was directly determined using flame photometry and converted to obtain the available potassium content of the soil.

### 3.6. Determination of Tea Quality Index Content

The collected leaves of 44 different varieties of tea trees were bioinactivated at 105 °C for 15 min, dried at 80 °C until constant weight, ground, and passed through a 60-mesh sieve for the determination of quality indexes. The quality indexes of tea leaves were mainly determined for tea polyphenols, theanine, and caffeine, and three independent replicates were determined for each quality index. Among them, the tea polyphenol content was determined using UV spectrophotometry with reference to the National Standard of the People’s Republic of China, GB/T 8313-2018 [79]. Briefly, 1 g of tea sample was weighed and 5 mL of 70% methanol solution was added and kept warm at 70 °C for 10 min. The above solution was centrifuged, 1 mL of supernatant was taken, and 5 mL of Folin–Ciocalteu reagent was added. After 5 min of reaction, 4 mL of 7.5% Na_2_CO_3_ solution was added, and the absorbance was measured at 765 nm after 60 min of standing. Tea polyphenols were quantified with reference to gallic acid as a standard curve. Theanine content was determined using high-performance liquid chromatography (HPLC) with reference to the National Standard of the People’s Republic of China, GB/T 23193-2017 [80]. Briefly, 1 g of tea sample was weighed, added to 100 mL of boiling distilled water, subjected to a water bath at 100 °C for 30 min, and was then filtered and fixed. The above filtrate was passed through a 0.45 μm filter membrane and subjected to high-performance liquid chromatography (HPLC) for the determination and quantification of theanine with reference to the standard curve of theanine. Caffeine content was determined using UV spectrophotometry with reference to the National Standard of the People’s Republic of China, GB/T 8312-2013 [81]. Briefly, 3 g of tea sample was weighed, added to 450 mL of boiling distilled water, and kept in a water bath at 100 °C for 45 min, before being filtered and concentrated. A total of 10 mL of filtrate was taken, 4 mL of 0.01 mol/L hydrochloric acid was added, the volume was then fixed to 100 mL, and was filtrated for standing, and the absorbance was measured at 274 nm to quantify caffeine with reference to the standard curve of caffeine.

### 3.7. Statistical Analysis

Excel 2017 software was used to preprocess raw data with routine graphic production, including calculation of mean, variance, etc., as well as production of stacked and bar charts. Rstudio software (R version 4.2.3) was used to produce K-mean clustering maps (R library was vegan 2.6.4 and fpc 2.2.10), Bray–Curtis heat maps (R library was pheatmap 1.0.12), the orthogonal partial least squares-discriminate analysis model (OPLS-DA, package used for this was ropls and mixOmics), box plots (R library was gghalves 0.1.4), scatter plots (R library was ggplot 2 3.4.0), redundancy analysis (R library was vegan 2.6.4), interaction network diagrams (R library was linkET 0.0.7.1), and the partial least square—structural equation model (PLS-SEM, library was plspm version 0.5.0).

## 4. Conclusions

In this study, we analyzed the effects of different varieties of tea tree germplasm resources on the structure of bacterial communities from the perspective of rhizosphere soil bacteria, and explored whether the memory behavior of tea trees towards rhizosphere soil bacteria affects the recruitment and aggregation capacity of bacterial communities, which in turn affects soil nutrient transformations and alters tea leaf quality. The results of the study showed that the diversity and abundance of rhizosphere soil bacterial communities did differ significantly among the 44 tea tree varieties, but the dominant bacterial populations in their rhizosphere soils were consistent. It is evident that the ecological memory of tea trees for rhizosphere soil bacteria determines the consistency of the bacterial communities they prefer to recruit and aggregate. Secondly, we found that the different varieties of tea trees could be effectively distinguished from each other in terms of their bacterial communities (group A and group B), and that the characteristic bacteria that distinguished them were from 23 genera, and that the key to distinguishing them was a significant difference in the abundance of the characteristic bacteria. Our quantitative validation of the characteristic bacteria using *q*RT-PCR also confirmed the results. In addition, we found that changes in the number of these characteristic bacteria affected the available nutrient content of the soil, which in turn affected tea quality. Based on the content of soil available nutrients and tea quality indexes, five machine learning methods were used to simulate the classification of 44 different varieties of tea trees, and it was very surprising that the results of their simulated classifications were consistent with those of the simulated classifications based on the bacterial community, and showed that the content of soil available nutrients and the content of tea quality indexes of group A were significantly higher than those of group B. PLS-SEM model analysis also demonstrated that there was a positive effect of the number of characteristic bacteria on the content of soil available nutrients, and a positive effect of the content of soil available nutrients on the content of tea quality indexes. This study shows that tea trees have an ecological memory of rhizosphere soil bacteria, and different varieties of tea trees can be differentiated according to their quality from the perspective of rhizosphere soil bacteria. The memory behavior of different tea tree varieties towards rhizosphere soil bacteria determines their ability to recruit and aggregate rhizosphere soil bacterial communities, and the strength of this ability increases the availability of rhizosphere soil nutrients and determines tea quality. This study proposes a new explanatory approach to the previous view that differences in plant germplasm resource quality may be due to varietal differences, i.e., plant ecological memory for rhizosphere soil bacteria is one of the most important factors contributing to differences in plant germplasm resource quality. This study provides an important research basis for the future regulation of plant cultivation through bacterial communities, thereby improving plant quality. Of course, differences in soil physicochemical indexes may also lead to changes in microbial communities, which need to be further explored in future studies.

## Figures and Tables

**Figure 1 plants-13-01686-f001:**
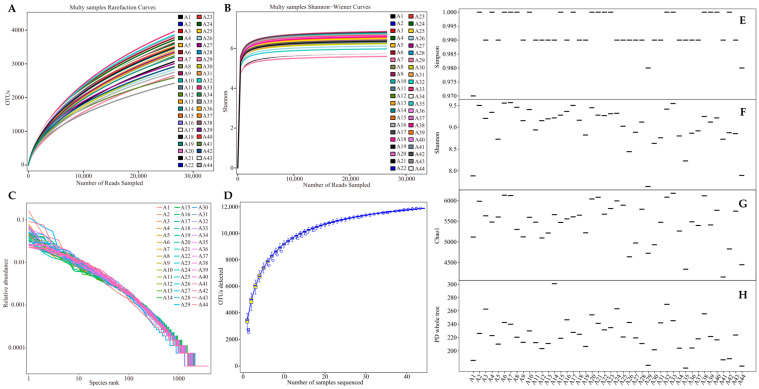
OTUs analysis of rhizosphere soil bacteria of 44 tea tree germplasm resources. A1~A44 indicate different varieties of tea trees. (**A**) Rarefaction curve analysis of rhizosphere soil bacterial OTUs; (**B**) Shannon–Wiener curve analysis of rhizosphere soil bacterial diversity; (**C**) rank-abundance curve plot of rhizosphere soil bacteria; (**D**) species accumulation curve plot of rhizosphere soil bacteria; (**E**) Simpson index analysis of rhizosphere soil bacteria; (**F**) Shannon index analysis of rhizosphere soil bacteria; (**G**) Chao1 index analysis of rhizosphere soil bacteria; and (**H**) PD whole tree index analysis of rhizosphere soil bacteria.

**Figure 2 plants-13-01686-f002:**
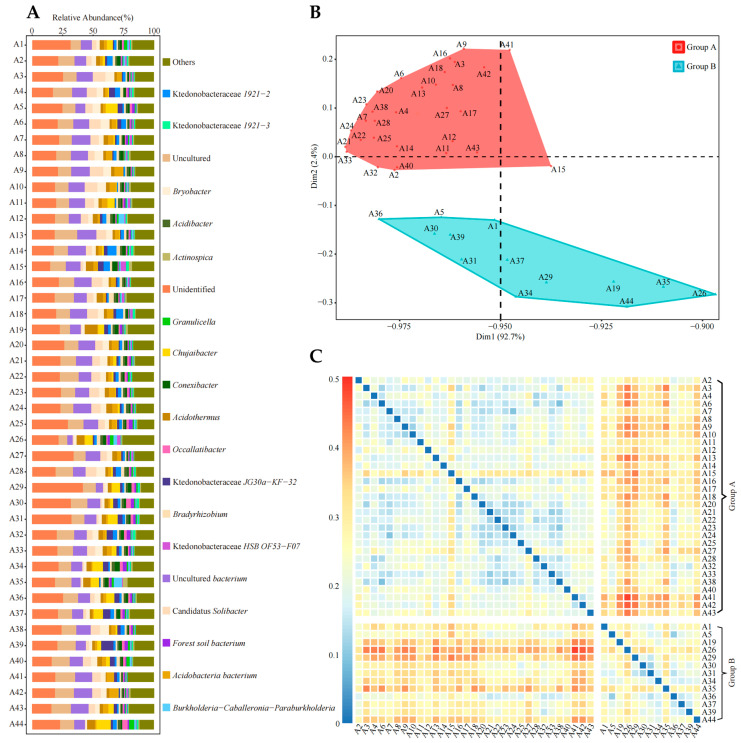
Rhizosphere soil bacterial abundance and classification analysis of 44 tea tree germplasm resources. A1~A44 indicate different varieties of tea trees. (**A**) Rhizosphere soil bacterial abundance analysis (the figure shows information on genera with a relative abundance of 1% or more); (**B**) K-mean cluster analysis of different tea tree germplasm resources based on rhizosphere soil bacterial abundance; (**C**) Bray–Curtis heat map analysis of bacterial abundance in group A and group B after K-mean clustering.

**Figure 3 plants-13-01686-f003:**
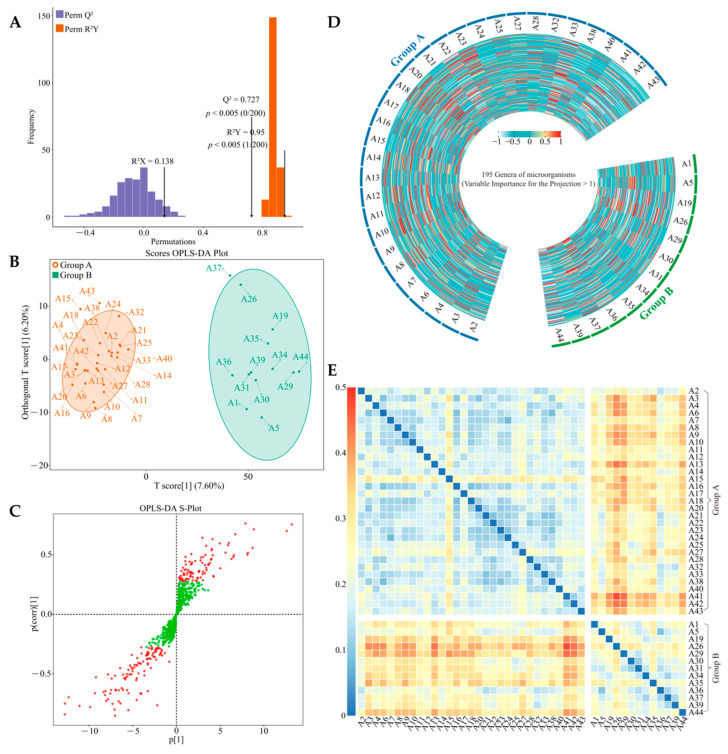
Screening of key bacteria after classification of 44 tea tree germplasm resources. A1~A44 indicate different varieties of tea trees. (**A**) OPLS-DA model test plot for key differential bacterial screening for group A and group B; (**B**) OPLS-DA model score plot analysis for group A and group B; (**C**) S-Plot analysis of the OPLS-DA model for group A and group B (Red dots indicate fungi with VIP > 1 and green dots indicate fungi with VIP < 1); (**D**) heatmap analysis of the abundance of key differential bacteria between group A and group B obtained from the OPLS-DA model; (**E**) Bray–Curtis heatmap analysis of group A and group B with the abundance of 195 key differential bacterial genera.

**Figure 4 plants-13-01686-f004:**
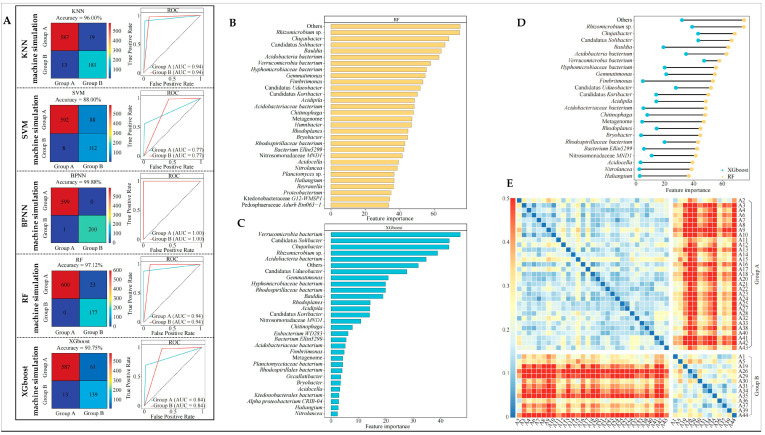
Machine deep learning simulation based on key bacterial abundance in 195 genera to validate classification accuracy and screening to obtain characteristic bacterial genera. A1~A44 indicate different varieties of tea trees. (**A**) Five different machine learning methods to validate the accuracy of key bacteria classification; (**B**) using RF to obtain the importance value of key bacterial genera in classification; (**C**) importance values of key bacterial genera in the classification were obtained using XGboost; (**D**) importance values of 23 characteristic bacterial genera ranked in the top 30 and shared by RF and XGboost; (**E**) Bray–Curtis heatmap analysis of the abundance of the 23 characteristic bacterial genera in group A and group B.

**Figure 5 plants-13-01686-f005:**
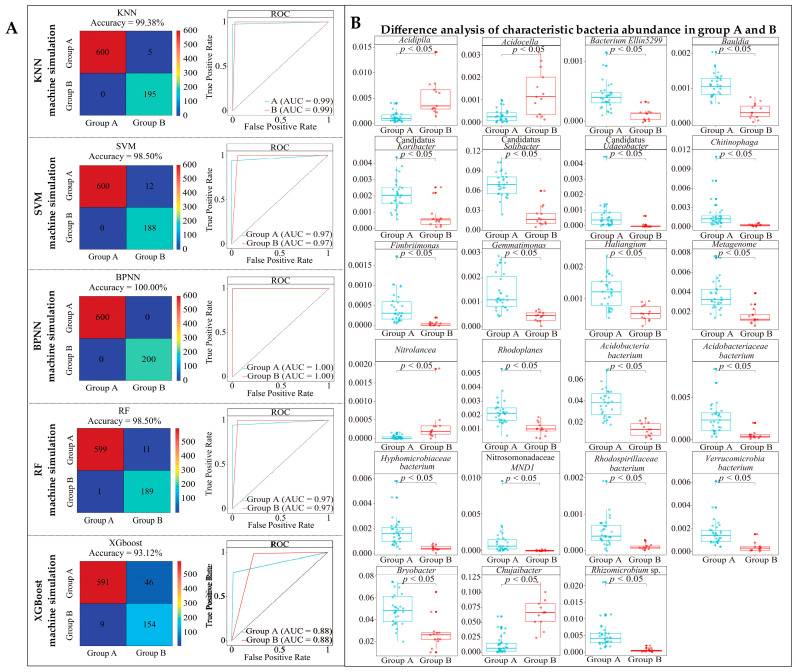
Machine deep learning based on the abundance of 23 characteristic bacterial genera to validate classification accuracy and abundance analysis. (**A**) Accuracy of five different machine learning methods to validate the classification of characteristic bacterial genera. (**B**) Difference analysis of abundance of characteristic bacterial genera in group A and group B.

**Figure 6 plants-13-01686-f006:**
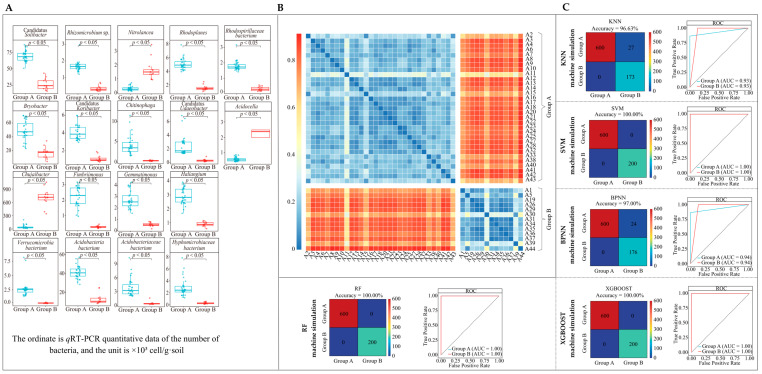
*q*RT-PCR analysis of rhizosphere soil characteristic bacterial genera of tea tree and validation of the accuracy of their classification. (**A**) Box plots of *q*RT-PCR results for 18 characteristic bacterial genera in group A and group B tea tree germplasm resources; (**B**) Bray–Curtis heatmap of the number of characteristic bacterial genera in group A and group B based on *q*RT-PCR results of 18 characteristic bacterial genera; (**C**) five different machine learning methods to validate the classification accuracy of 18 characteristic bacterial genera.

**Figure 7 plants-13-01686-f007:**
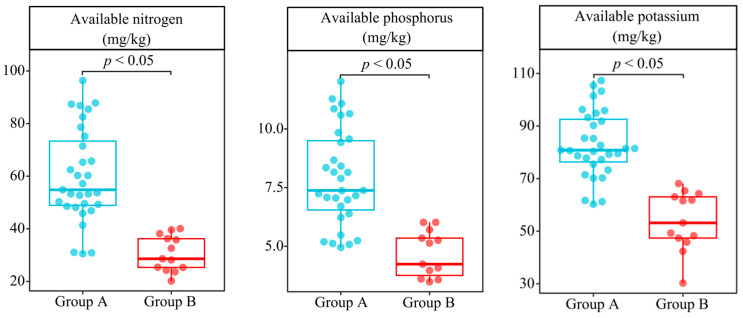
Analysis of rhizosphere soil available nutrient content of 44 tea tree germplasm resources.

**Figure 8 plants-13-01686-f008:**
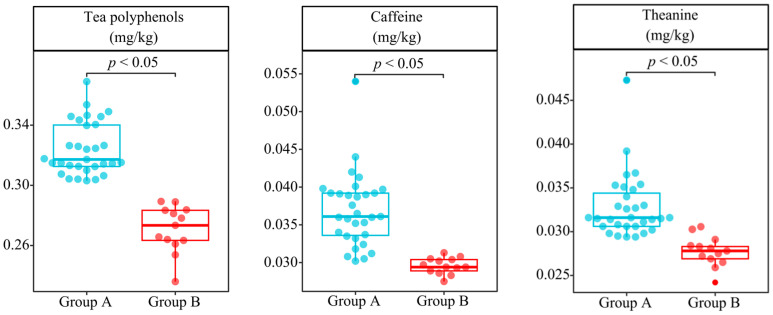
Leaf quality index content analysis of 44 tea tree germplasm resources.

**Figure 9 plants-13-01686-f009:**
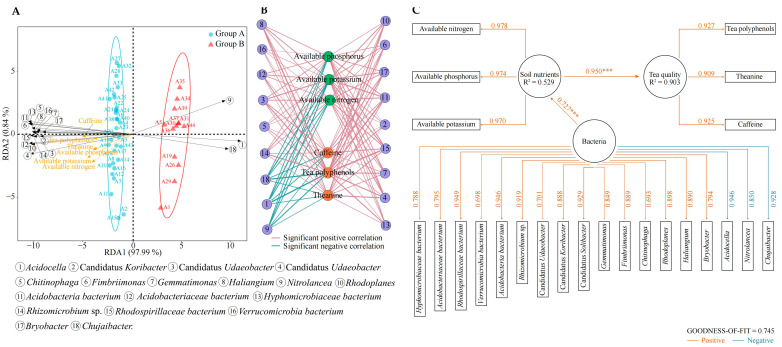
Interaction relationship analysis between soil characteristic microorganisms of tea tree rhizosphere and soil available nutrients and tea quality indexes. (**A**) Redundancy analysis of soil characteristic microorganisms with soil available nutrients and tea quality indexes; (**B**) correlation network analysis of soil characteristic microorganisms with soil available nutrients and tea quality indexes; (**C**) PLS-SEM model analysis of soil characteristic microorganisms with soil available nutrients and tea quality indexes (*** indicates significance at the *p* < 0.001 level).

**Figure 10 plants-13-01686-f010:**
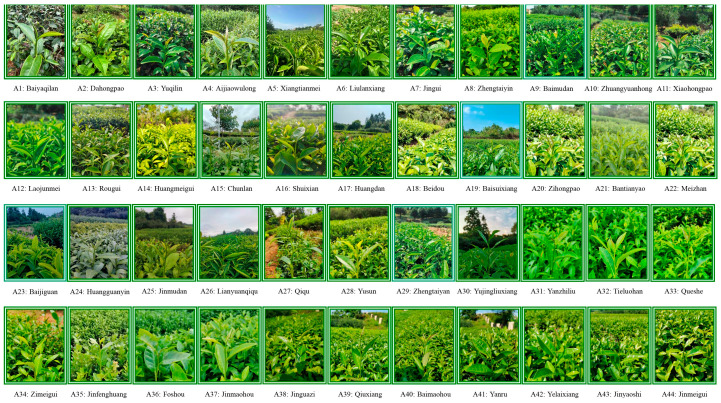
Photographs of 44 different varieties of tea trees selected for the experiment.

**Figure 11 plants-13-01686-f011:**
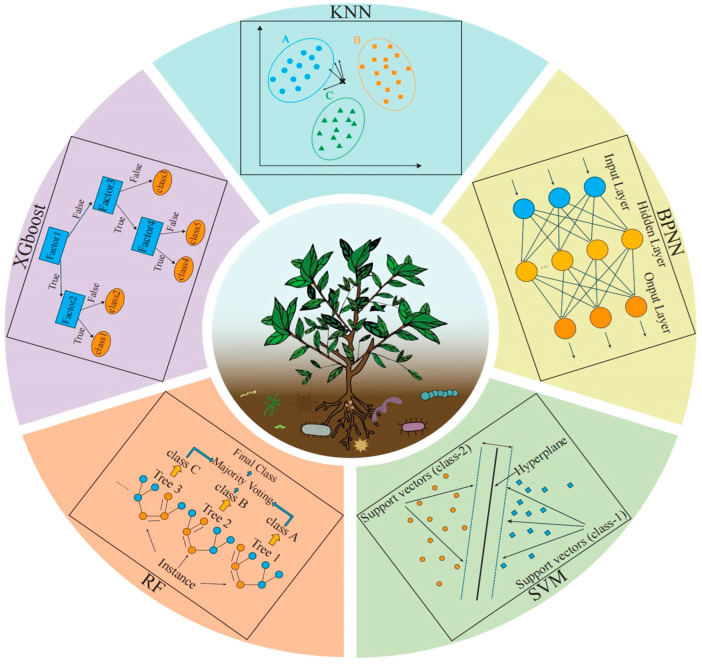
Basic principle diagram of five machine learning algorithms.

**Table 1 plants-13-01686-t001:** Numbers and names of 44 different varieties of tea trees selected for the experiment.

No.	*Camellia sinensis* Variety	No.	*Camellia sinensis* Variety	No.	*Camellia sinensis* Variety	No.	*Camellia sinensis* Variety
A1	Baiyaqilan	A12	Laojunmei	A23	Baijiguan	A34	Zimeigui
A2	Dahongpao	A13	Rougui	A24	Huangguanyin	A35	Jinfenghuang
A3	Yuqilin	A14	Huangmeigui	A25	Jinmudan	A36	Foshou
A4	Aijiaowulong	A15	Chunlan	A26	Lianyuanqiqu	A37	Jinmaohou
A5	Xiangtianmei	A16	Shuixian	A27	Qiqu	A38	Jinguazi
A6	Liulanxiang	A17	Huangdan	A28	Yusun	A39	Qiuxiang
A7	Jingui	A18	Beidou	A29	Zhengtaiyan	A40	Baimaohou
A8	Zhengtaiyin	A19	Baisuixiang	A30	Yujingliuxiang	A41	Yanru
A9	Baimudan	A20	Zihongpao	A31	Yanzhiliu	A42	Yelaixiang
A10	Zhuangyuanhong	A21	Bantianyao	A32	Tieluohan	A43	Jinyaoshi
A11	Xiaohongpao	A22	Meizhan	A33	Queshe	A44	Jinmeigui

## Data Availability

All raw reads were deposited in the National Center for Biotechnology Information (NCBI, http://www.ncbi.nlm.nih.gov, accessed on 16 May 2023) Short Read Archive (SRA) database under accession number PRJNA973728.

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
