# Peer review of "Recruitment and Aggregation Capacity of Tea Trees to Rhizosphere Soil Characteristic Bacteria Affects the Quality of Tea Leaves"

_plants, 2024, doi:10.3390/plants13121686_

Round 1

Reviewer 1 Report

Comments and Suggestions for Authors

This manuscript describes the bacterial community structure in soil of several tea (Camellia sinensis) plantations in China and its relation to the quality of tea plants. The major methods used were 16S rRNA gene amplicon sequencing, real-time quantitative PCR, machine deep learning of the phylogenetic data, and chemical characterization of the tea plants. Based on these approaches, the study provides some interesting information on the relationship between the soil microbiota and the quality of different varieties of tea tree germplasm resources, where the plants are categorized into the major two groups.  Despite this, there are major drawbacks and limitations in this study. The authors should carefully consider these limitations and thereby redraw their conclusions. I have the following major critical comments and suggestions for author’s considerations.

1. The title of the paper is a leap and should be revised to be more in line with the content of the paper.

2. Archaea as well as bacteria may play important roles in soil, especially in acidic soil of tea plantation. For some reason, however, archaea is ignored in this study.

3. This paper lacks information on the total bacterial counts in the soil studied. This information is important to know actual populations of different phylotypes identified by 16S rRNA gene amplicon sequencing.

4. The description of the RT-PCR method is incomplete (L686-L694). Was this PCR targeting the 16S rRNA gene? If so, it should be explicitly stated. Also, what commercial kit did the authors used to do this?  How did they make the standard curve for quantitation? Since the copy number of the 16S rRNA gene varies among different bacterial species, it is not possible to obtain a quantitative value equivalent to the number of bacterial cells directly from the amplified amount unless the gene copy number is corrected (L334-425). A standard curve should be prepared to determine the number of cells with a bacterial species that has a copy number close to the average copy number of all bacterial species. Alternatively, the total bacterial count by DNA staining and the relative ratio of amplicons with each specific primer set to the amplified amount with a universal primer set can be used to calculate each targeted population.

5. The phylogenetic and taxonomic descriptions of bacteria for 16S rRNA gene amplicon data are too poor to understand their diversity in the studied soils, and need to be totally revised.  
5-1 The sequence similarity level to assign the phylotypes to genera should be clearly shown. The authors should not confuse a mere phylogenetic group (phylotype) with a genus officially defined. Basically, any of the bacterial names that do not appear as the generic name in the LPSN (https://www.bacterio.net/) should be described as closest relatives (with database accession numbers), not as generic names.
5-2 For example, the terms 1921-2, 1921-3, “unidentified”, G30a-KF-32, HSB OF53-F07, “uncultured bacterium” (L189-193), uncultured Acidobacteria bacterium, uncultured Verrucomicrobia bacterium (L285-386), and so on do not make sense as they stand. At least, the database accession number of sequences and/or phylum names for these organisms identified as the closest relative should be given to make their identity clear.
5-3 The scientific name for Candidatus should be of the roman type, not italic.

6. Information on some important physicochemical parameters of the studied soils, e.g., pH and conductivity, is lacking. Generally tea plantation soils become acidic due to excess nitrogenous fertilizer application.  In fact, the authors identified the phylotypes that correspond to the acidic genera such as Acidibacter, Acidothermus, and those of the phylum Acidobacteriota. Discussion on the relationship between the acidity of the soil studied and the bacterial community structure should be essential.

7. Results of machine deep learning of the phylotypes identified (L259-298) should be carefully rechecked. Although the authors mentioned “Machine deep learning validates 195 key bacterial genera, they did not provide evidence that all these corresponded to the genera (please find my comments 5).

8. Figure 2A, Figure 4B-D, Figure 5B, C, Figure 6A, and Table S4 include meaningless bacterial names. The bacterial names should be revised as the genera assigned at a sequence similarity level or closest relatives with database accession numbers.

9. Figure 5B can be deleted. This is the case in the left part of Figure 6A.

Author Response

Comments and Suggestions for Authors

This manuscript describes the bacterial community structure in soil of several tea (Camellia sinensis) plantations in China and its relation to the quality of tea plants. The major methods used were 16S rRNA gene amplicon sequencing, real-time quantitative PCR, machine deep learning of the phylogenetic data, and chemical characterization of the tea plants. Based on these approaches, the study provides some interesting information on the relationship between the soil microbiota and the quality of different varieties of tea tree germplasm resources, where the plants are categorized into the major two groups.  Despite this, there are major drawbacks and limitations in this study. The authors should carefully consider these limitations and thereby redraw their conclusions. I have the following major critical comments and suggestions for author’s considerations.

1. The title of the paper is a leap and should be revised to be more in line with the content of the paper.
A: Thank you to the reviewer. The authors have revised the title.

2. Archaea as well as bacteria may play important roles in soil, especially in acidic soil of tea plantation. For some reason, however, archaea is ignored in this study.
A: Many thanks to the reviewers. Yes, soil contains not only bacteria but also archaea, fungi, and viruses. They are all likely to have an impact on the growth and quality of the tea tree. In this manuscript, the authors focus on the research of bacterial perspective. In the future, the authors will further delve into the effects of changes in other factors in the soil on the growth of tea trees. Thank you very much for your suggestions.

3. This paper lacks information on the total bacterial counts in the soil studied. This information is important to know actual populations of different phylotypes identified by 16S rRNA gene amplicon sequencing.
A: Many thanks to the reviewers. All the tea trees selected by the authors were planted in the same germplasm resource nursery, and the initial state of their soils is basically the same.

4. The description of the RT-PCR method is incomplete (L686-L694). Was this PCR targeting the 16S rRNA gene? If so, it should be explicitly stated. Also, what commercial kit did the authors used to do this?  How did they make the standard curve for quantitation? Since the copy number of the 16S rRNA gene varies among different bacterial species, it is not possible to obtain a quantitative value equivalent to the number of bacterial cells directly from the amplified amount unless the gene copy number is corrected (L334-425). A standard curve should be prepared to determine the number of cells with a bacterial species that has a copy number close to the average copy number of all bacterial species. Alternatively, the total bacterial count by DNA staining and the relative ratio of amplicons with each specific primer set to the amplified amount with a universal primer set can be used to calculate each targeted population.
A: Many thanks to the reviewers. First, the authors targeted the conserved sequence of the 16S rRNA gene of each microorganism to design primers before PCR amplification. Second, the authors made standard curves with known concentrations of plasmid and then quantified each sample based on the standard curves. The authors have also made an addition to materials and methods to explain the above. Many thanks to the reviewers for their suggestions.

5. The phylogenetic and taxonomic descriptions of bacteria for 16S rRNA gene amplicon data are too poor to understand their diversity in the studied soils, and need to be totally revised.  
5-1The sequence similarity level to assign the phylotypes to genera should be clearly shown. The authors should not confuse a mere phylogenetic group (phylotype) with a genus officially defined. Basically, any of the bacterial names that do not appear as the generic name in the LPSN (https://www.bacterio.net/) should be described as closest relatives (with database accession numbers), not as generic names.

A: Many thanks to the reviewers. The authors have done a thorough check and revision.

5-2 For example, the terms 1921-2, 1921-3, “unidentified”, G30a-KF-32, HSB OF53-F07, “uncultured bacterium” (L189-193), uncultured Acidobacteria bacterium, uncultured Verrucomicrobia bacterium (L285-386), and so on do not make sense as they stand. At least, the database accession number of sequences and/or phylum names for these organisms identified as the closest relative should be given to make their identity clear.

A: Thanks to the reviewers. The authors have thoroughly checked the text and images and made appropriate changes.

5-3 The scientific name for Candidatus should be of the roman type, not italic.
A: Thanks to the reviewers. The authors have thoroughly checked and revised it.

6. Information on some important physicochemical parameters of the studied soils, e.g., pH and conductivity, is lacking. Generally tea plantation soils become acidic due to excess nitrogenous fertilizer application.  In fact, the authors identified the phylotypes that correspond to the acidic genera such as AcidibacterAcidothermus, and those of the phylum Acidobacteriota. Discussion on the relationship between the acidity of the soil studied and the bacterial community structure should be essential.
A: Thanks to the reviewers. Many physicochemical indexes in the soil do affect the microorganisms, especially after excessive application of nitrogen fertilizers, the soil becomes acidified and definitely affects the microorganisms. Meanwhile, this study focused on exploring the microbial distinctions between different tea germplasm resources. Therefore, the effect of soil physicochemical indexes on microorganisms was not analyzed. Certainly, the expert's advice is excellent. The authors will focus on the effect of different soil physicochemical indexes on microorganisms in our subsequent studies. Thanks again to the reviewing experts for their suggestions.

7. Results of machine deep learning of the phylotypes identified (L259-298) should be carefully rechecked. Although the authors mentioned “Machine deep learning validates 195 key bacterial genera, they did not provide evidence that all these corresponded to the genera (please find my comments 5).

A: Thanks to the reviewers for their suggestions. Perhaps the authors did not state clearly, and the authors found that tea tree germplasm resources could be categorized into two groups (Group A and Group B). Therefore, the authors used OPLS-DA analysis to find 195 genera of key bacteria distinguishing between Group A and Group B. The authors further used different types of machine learning, including supervised and unsupervised machine learning, to analyze whether 195 bacterial genera could effectively distinguish Group A from Group B. The results showed that 195 bacterial genera were indeed effective in differentiating between Group A and Group B.

Second, the authors screened for key bacterial genera to distinguish between Group A and Group B from the bacteria obtained, whereas the genera listed in Fig. 2 were only those with abundance greater than 1%, and those bacterial genera greater than 1% did not necessarily distinguish between Group A and Group B.

8. Figure 2A, Figure 4B-D, Figure 5B, C, Figure 6A, and Table S4 include meaningless bacterial names. The bacterial names should be revised as the genera assigned at a sequence similarity level or closest relatives with database accession numbers.
A: Thank you to the reviewers. The authors have carefully revised the graphics as required, and have also scrutinized and revised the text throughout. Thanks again to the reviewing experts.

9. Figure 5B can be deleted. This is the case in the left part of Figure 6A.

A: Thank you to the reviewers. The authors have removed unnecessary graphics and reformatted the manuscript.

Reviewer 2 Report

Comments and Suggestions for Authors

The authors never mention that plants are not single organisms, but part of a holobiontic biosystem, a community plant+(micro)organisms in phylosphere and rhizosphere, in endosphere and ectosphere. Knowledge is well established, and is common part of the understanding of ecosystems. Microbiota as part of the holobiont are important plant partners, so in case of nutrient stress, resistance to biotic and abiotic stresses, so drought resistance, metal tolerance, etc., and more. Microbiota is modulated by plants for adaptation to local soil conditions, for optimal mobilization and uptake of resources. It is well known, that in a soil with high nitrogen supply, leguminous plants build up less nodules with N2-fixing rhizobia, in soil with good nutrient supply, plant feed less mycorrhiza, so that less resources of the plant (photosynthates) are used for external needs, and more resources should be investigated in plant growth etc… Such tradeoff’s are developed in literature.

Also plant breeding often considers the assemblies with microbiota, partly endophytic bacteria.

So the authors should consider the rules of holobiont biosystems in the validation of the results. I propose “major revision” and new reviewing of the improved version.

Two supplementary remarks

Line 686: „Based on the previous studies and analysis” - Another study? Other materials?

Line 776. “the strength of this ability determines the content of available nutrients in the rhizosphere soil“.

I wonder this conclusion. Plant holobiotic organisms may increase nutrients availability, by many well-known mechanisms, but determine not per se the content of available nutrients..

I propose for future research to test some of the 44 varieties of tea tree also in soil with differing fertility, differing properties, for verification of the stability of the characteristic microorganism populations supported by plants, verification of the hypothesis “Ecological Memory”

Author Response

Comments and Suggestions for Authors

The authors never mention that plants are not single organisms, but part of a holobiontic biosystem, a community plant+(micro)organisms in phylosphere and rhizosphere, in endosphere and ectosphere. Knowledge is well established, and is common part of the understanding of ecosystems. Microbiota as part of the holobiont are important plant partners, so in case of nutrient stress, resistance to biotic and abiotic stresses, so drought resistance, metal tolerance, etc., and more. Microbiota is modulated by plants for adaptation to local soil conditions, for optimal mobilization and uptake of resources. It is well known, that in a soil with high nitrogen supply, leguminous plants build up less nodules with N2-fixing rhizobia, in soil with good nutrient supply, plant feed less mycorrhiza, so that less resources of the plant (photosynthates) are used for external needs, and more resources should be investigated in plant growth etc… Such tradeoff’s are developed in literature.

Also plant breeding often considers the assemblies with microbiota, partly endophytic bacteria.

So the authors should consider the rules of holobiont biosystems in the validation of the results. I propose “major revision” and new reviewing of the improved version.

A: Many thanks to the reviewer for the suggestion. Yes, the interaction between plants and microorganisms during cultivation is a holistic concept. Changes in environmental factors alter the structure of the soil microbial community and affect plant growth. An equilibrium is formed after this interaction. This manuscript focused on the differences in rhizosphere soil bacteria among different tea germplasm resources under the same environmental conditions, and tried to find characteristic bacteria and analyze their effects on soil available nutrients and tea quality. Based on the suggestions of the reviewers, the authors have made appropriate changes to the manuscript. Hopefully, it will meet the requirements.

Two supplementary remarks

Line 686: „Based on the previous studies and analysis” - Another study? Other materials?

A: Thank you to the reviewer. The authors did not express themselves clearly. What the authors were trying to express was that the results obtained in the above part of the study were what was studied in this manuscript not another study.

Line 776. “the strength of this ability determines the content of available nutrients in the rhizosphere soil“.

I wonder this conclusion. Plant holobiotic organisms may increase nutrients availability, by many well-known mechanisms, but determine not per se the content of available nutrients.

A: Thank you to the reviewer. Yes, it should be to improve nutrient availability. Thank you very much for your suggestion and the authors have revised the formulation of the statement.

I propose for future research to test some of the 44 varieties of tea tree also in soil with differing fertility, differing properties, for verification of the stability of the characteristic microorganism populations supported by plants, verification of the hypothesis “Ecological Memory”。

A: Many thanks to the reviewers for their suggestions. More varieties of tea tree were selected in this study, and this study initially explored the differences among tea tree germplasm resources. The authors and their team are already working on, the effect of different fertilization practices, different soil acidity, etc., on the rhizosphere characteristic microorganisms of tea trees. Thanks again to the reviewers for their suggestions.

Round 2

Reviewer 1 Report

Comments and Suggestions for Authors

As I mentioned and suggest for considerations before, there are several limitations in this study. The manuscript has been partially revised, but there is no mention of any of these limitations. The following limitations should be fully stated in the discussion.
1. Lack of quantitative values for microbial abundance.
2. No indication of soil pH or other physicochemical parameters
3. No discussion of tea plantation soil acidity and its effect on microbial community structure
4. No study of archaeal distribution and significance (e.g., their roles in nitrification in acidic soil)
5. Possible bias in the results due to these limitations

The phylogenetic and taxonomic descriptions of bacteria on 16S rRNA gene amplicon data have not yet been fully revised. As I pointed out, the terms “uncultured” (L189), “unidentified” (L190), “uncultured bacterium” (L192), “Forest soil bacterium” (L192), Acidobacteria bacterium (L193), Burkholderia-Caballeronia-Paraburkholderia (L193), and so on do not make sense as they stand. Nobody can understand what these organisms are. To make the identity of all uncultured bacterial clones described, the database clone names with accession numbers should be indicated where they first appear in the text. For example, the accession number for Ktedonobacteraceae JG30a−KF−32 is AJ536876, and this uncultured clone can be described as follows: uncultured Ktedonobacteraceae bacterium JG30a−KF−32 (accession number: AJ536876). Also, all uncultured clone names the authors mentioned should be carefully rechecked: Acidobacteria bacterium → uncultured acidobacterium (if a clone name is given in database, please specify it), Ktedonobacteraceae 1921−2 → uncultured Ktedonobacteraceae bacterium 1921−2, and so on.

Describing “21 genera” (L188) is not correct, but this should read “phylotypes”. Therefore, for example, the sentence "The results showed (Figure 2A) that a total of 745 genera of bacteria were identified in the rhizosphere soil of tea tree, of which 21 genera with a relative abundance of more than 1%..." should be rewritten to make sense..

I show an example for this:
"Using the RDP classifier, the 147,765 OTUs obtained were classified into 745 phylotypes that could be assigned with members of the established bacterial genera and the previously reported uncultured clones considered independent at the genus level. Out of the 745 phylotypes, the following 20 accounted for more than 1% of the total abundance: uncultured Ktedonobacteraceae bacterium 1921−2 (database accession number,xxxxxxx), uncultured Ktedonobacteraceae bacterium 1921−3 (xxxxxxx), ......".

If the terms "Others", "unidentified", and "uncultured" mean a mixture of different phylotypes at the generic level, then they should be all eliminated in the counting of the number of phylotypes.

Author Response

Comments and Suggestions for Authors

As I mentioned and suggest for considerations before, there are several limitations in this study. The manuscript has been partially revised, but there is no mention of any of these limitations. The following limitations should be fully stated in the discussion.
1. Lack of quantitative values for microbial abundance.

A: Thanks to the reviewers. Regarding the analysis of microbial amplicons, the first step was to discriminate by abundance, on the basis of which the characteristic microorganisms were obtained. The authors then performed quantitative measurements for the screened characteristic microorganisms.

2. No indication of soil pH or other physicochemical parameters

A: Thank you to the reviewers. The authors have provided additional information in the conclusion.

3. No discussion of tea plantation soil acidity and its effect on microbial community structure

A: Thank you to the reviewers. Changes in soil acidity do have an impact on microbial community structure. First, the tea tree germplasm resources in this study came from one germplasm resource nursery, planted in the same plot with similar soil properties. Second, this study mainly explored the differences between different tea tree germplasm resources. Therefore, the acidity of tea plantation soil and its effect on microbial community structure was not discussed.

4. No study of archaeal distribution and significance (e.g., their roles in nitrification in acidic soil)

A: Thank you to the reviewers.  Archaea are also a large taxon that needs to be studied and analyzed specifically. The study focused on bacterial communities. For the changes in the archaea, authors would subsequently continue to study it in depth.

5. Possible bias in the results due to these limitations
A: Thanks to the reviewers. This study was conducted to investigate the bacterial communities in the rhizosphere soil of different tea tree germplasm resources from the same plot. Of course, there are some deficiencies in some aspects. Thanks to the experts for their comments.

The phylogenetic and taxonomic descriptions of bacteria on 16S rRNA gene amplicon data have not yet been fully revised. As I pointed out, the terms “uncultured” (L189), “unidentified” (L190), “uncultured bacterium” (L192), “Forest soil bacterium” (L192), Acidobacteria bacterium (L193), Burkholderia-Caballeronia-Paraburkholderia (L193), and so on do not make sense as they stand. Nobody can understand what these organisms are. To make the identity of all uncultured bacterial clones described, the database clone names with accession numbers should be indicated where they first appear in the text. For example, the accession number for Ktedonobacteraceae JG30a−KF−32 is AJ536876, and this uncultured clone can be described as follows: uncultured Ktedonobacteraceae bacterium JG30a−KF−32 (accession number: AJ536876). Also, all uncultured clone names the authors mentioned should be carefully rechecked: Acidobacteria bacterium → uncultured acidobacterium (if a clone name is given in database, please specify it), Ktedonobacteraceae 1921−2 → uncultured Ktedonobacteraceae bacterium 1921−2, and so on.

A: Thanks to the reviewers. Amplicon sequencing technique was used in this study and the sequenced results were categorized and counted according to genus. Some genera may contain a large number of species, and the accession number corresponds to each species, so it is not possible to supplement the accession number after the genus. In addition, in terms of nomenclature, a large number of scholars first categorized the results after sequencing the amplicons according to the genus and then named them after the genus, containing “uncultured”, “unidentified”, “uncultured bacterium”, “Forest soil bacterium”, Acidobacteria bacterium, Burkholderia-Caballeronia-Paraburkholderia, Ktedonobacteraceae 1921−2, Ktedonobacteraceae 1921−3.(Nature, Https://doi.org/10.1038/s41467-023-40810-z; Applied Geochemistry, https://doi.org/10.1016/j.apgeochem.2023.105779; Science Advances, https://doi.org/10.1126/sciadv.adk6295; Science Advances, https://doi.org/10.1126/sciadv.aba8555)

Describing “21 genera” (L188) is not correct, but this should read “phylotypes”. Therefore, for example, the sentence "The results showed (Figure 2A) that a total of 745 genera of bacteria were identified in the rhizosphere soil of tea tree, of which 21 genera with a relative abundance of more than 1%..." should be rewritten to make sense..
I show an example for this:
"Using the RDP classifier, the 147,765 OTUs obtained were classified into 745 phylotypes that could be assigned with members of the established bacterial genera and the previously reported uncultured clones considered independent at the genus level. Out of the 745 phylotypes, the following 20 accounted for more than 1% of the total abundance: uncultured Ktedonobacteraceae bacterium 1921−2 (database accession number,xxxxxxx), uncultured Ktedonobacteraceae bacterium 1921−3 (xxxxxxx), ......".

A: Thank you to the reviewers. The authors have made some revisions.

If the terms "Others", "unidentified", and "uncultured" mean a mixture of different phylotypes at the generic level, then they should be all eliminated in the counting of the number of phylotypes.

A: Thank you to the reviewers. Your suggestions are much appreciated. On amplicon analysis, "Others", "unidentified", and "uncultured" represent different categories, which are usually retained by scholars during the underlying analysis of the data.

Reviewer 2 Report

Comments and Suggestions for Authors

I see no real improving of the manuscript and list here some comments and personal meanings issued from my own word and projects.

The editor may decide about the acceptance of the manuscript for PLANTS. I propose to reject the manuscript given its philosophy of “new”. Another wording and positioning of rhizosphere microbiology, relating plants as holistic biosystem with their microbiome, as a long-time known fact, should result in the acceptability of the manuscript.

 Line 77: replace routinely by often

Line 78: when studying plant-microbial community interactions by botanists. (ecologists are not overlooking this knowledge). Since decencies, ecologists ask the botanists to consider also the plant root in their approaches, and offer the roots in glass house experiments a real growth ecosystem (soil) instead of nutrient solution or sand with nutrients, of in high doses. Ecologists include root and soil, comprising soil organisms, in their research. Mycorrhiza research in only one example, also bacteriologists, rhizosphere microbiologists are other examples of microbiome interactions with plants over their roots, may be nutrients, may be stress, may be defense, may be N2-fixation. A burst was given in the 1990 as the access to molecular approaches started, with detailed descriptions of populations activated in rhizospheres. The development of proteomic and bioinformatics followed, resulting also in performed appropriate rhizospheric microbial analysis with new insights and understanding. The microbiome was not long a black box, analyzed by biomass quantifications and culture methods.

Line 108: The authors hypothesized that another memory behavior to prompt recruitment and aggregation of memory microbes in the tea tree rhizosphere to re-establish the memory microbial community structure- This recruitment depends about the ecology of the soil or rhizosphere zone, and only populations are activated, that provide specific needed service for the pant (p.e. mobilization of certain nutrients), as plant deliver as few energy and synthetase’s as possible out of the roots, these amounts being no longer disponible for own growth. So, given past approaches of the ecologists in knowledge about the extreme high microbial biodiversity in soils, the own hypothesis of the authors seems not to as new as postulated.

Line 807 This study provides an important research basis for the future regulation of plant cultivation through bacterial communities, thereby improving plant quality.  Given the future regulation etc: decencies again, this research was active, as described not only in view of mobilization of nutrients, but also for other aspects so the mechanical soil disaggregation per root-steered bacterial slimes to allow a better root penetration. More examples are accessible in literature published most before the “molecular thrust”.

Author Response

Comments and Suggestions for Authors

I see no real improving of the manuscript and list here some comments and personal meanings issued from my own word and projects.

The editor may decide about the acceptance of the manuscript for PLANTS. I propose to reject the manuscript given its philosophy of “new”. Another wording and positioning of rhizosphere microbiology, relating plants as holistic biosystem with their microbiome, as a long-time known fact, should result in the acceptability of the manuscript.

 Line 77: replace routinely by often

A: Thank you to the reviewers. The authors have made revisions.

Line 78: when studying plant-microbial community interactions by botanists. (ecologists are not overlooking this knowledge). Since decencies, ecologists ask the botanists to consider also the plant root in their approaches, and offer the roots in glass house experiments a real growth ecosystem (soil) instead of nutrient solution or sand with nutrients, of in high doses. Ecologists include root and soil, comprising soil organisms, in their research. Mycorrhiza research in only one example, also bacteriologists, rhizosphere microbiologists are other examples of microbiome interactions with plants over their roots, may be nutrients, may be stress, may be defense, may be N2-fixation. A burst was given in the 1990 as the access to molecular approaches started, with detailed descriptions of populations activated in rhizospheres. The development of proteomic and bioinformatics followed, resulting also in performed appropriate rhizospheric microbial analysis with new insights and understanding. The microbiome was not long a black box, analyzed by biomass quantifications and culture methods.

A: Thank you to the reviewers. The authors have misrepresented themselves here, and the relevant statement has been deleted. 

Line 108: The authors hypothesized that another memory behavior … to prompt recruitment and aggregation of memory microbes in the tea tree rhizosphere to re-establish the memory microbial community structure- This recruitment depends about the ecology of the soil or rhizosphere zone, and only populations are activated, that provide specific needed service for the pant (p.e. mobilization of certain nutrients), as plant deliver as few energy and synthetase’s as possible out of the roots, these amounts being no longer disponible for own growth. So, given past approaches of the ecologists in knowledge about the extreme high microbial biodiversity in soils, the own hypothesis of the authors seems not to as new as postulated.

A: Many thanks to the reviewers. The authors speculate that this behavior may exist in tea trees from the perspective of germplasm resources. Few studies have been reported on the characterization and differences in rhizosphere soil microbial diversity of tea tree germplasm resources. The study is somewhat innovative.

Line 807 “This study provides an important research basis for the future regulation of plant cultivation through bacterial communities, thereby improving plant quality.”  Given the future regulation etc: decencies again, this research was active, as described not only in view of mobilization of nutrients, but also for other aspects so the mechanical soil disaggregation per root-steered bacterial slimes to allow a better root penetration. More examples are accessible in literature published most before the “molecular thrust”.

A: Thank you to the reviewer. Your suggestion is very much appreciated. Regarding the research in this area, the author's team is also continuously exploring and revealing the mechanism in depth. Thanks again to the reviewer.

Round 3

Reviewer 1 Report

Comments and Suggestions for Authors

Again I strongly point out that the author is still totally wrong about the description of the phylotypes. To describe it only as "uncultured", "uncultured bacterium", "unidentified", "Forest soil bacterium", and so on without a scientific name makes no sense at all, unless there is an accession number that indicates the sequence identity. I mean that "unidentified Proteobacteria" is OK, but “unidentified” alone is meaningless. If these phylotypes could not be assigned to taxa with scientific names or given accession numbers for their sequences, they should be removed from the description of phylogenetic assignment and united with "others".

Author Response

Comments and Suggestions for Authors

Again I strongly point out that the author is still totally wrong about the description of the phylotypes. To describe it only as "uncultured", "uncultured bacterium", "unidentified", "Forest soil bacterium", and so on without a scientific name makes no sense at all, unless there is an accession number that indicates the sequence identity. I mean that "unidentified Proteobacteria" is OK, but “unidentified” alone is meaningless. If these phylotypes could not be assigned to taxa with scientific names or given accession numbers for their sequences, they should be removed from the description of phylogenetic assignment and united with "others".

A: Thank the reviewers for your suggestions. Based on the suggestions of the reviewers, the authors have categorized “uncultured”, “uncultured bacterium”, “unidentified”, and “Forest soil bacterium” as all others. Meanwhile, the author has revised the description and all the figures in the manuscript. I hope it meets the requirements. Thanks again to the reviewers.